# Land Use Hotspots of the Two Largest Landlocked Countries: Kazakhstan and Mongolia

Jing Yuan [1,*], Jiquan Chen [1,2], Pietro Sciusco [2], Venkatesh Kolluru [3], Sakshi Saraf [3], Ranjeet John [3] and Batkhishig Ochirbat [4]

1 Center for Global Change and Earth Observations, Michigan State University, East Lansing, MI 48823, USA; jqchen@msu.edu
2 Department of Geography, Environment & Spatial Sciences, Michigan State University, East Lansing, MI 48823, USA; sciuscop@msu.edu
3 Department of Biology and Department of Sustainability, University of South Dakota, Vermillion, SD 57069, USA; venkatesh.kolluru@coyotes.usd.edu (V.K.); sakshi.saraf@coyotes.usd.edu (S.S.); ranjeet.john@usd.edu (R.J.)
4 Institute of Geography, Mongolian Academy of Sciences, Ulaanbaatar 210620, Mongolia; batkhishig@gmail.com
* Correspondence: yuanji11@msu.edu

**Abstract:** As the two largest landlocked countries, Kazakhstan and Mongolia have similar biophysical conditions and socioeconomic roots in the former Soviet Union. Our objective is to investigate the direction, extent, and spatial variation of land cover change at three administrative levels over three decades (1990–2020). We selected three provinces from each country (Aktobe, Akmola, and Almaty province in Kazakhstan, and Arkhangai, Tov, and Dornod in Mongolia) to classify the land cover into forest, grassland, cropland, barren, and water. Altogether, 6964 Landsat images were used in pixel-based classification method with random forest model for image processing. Six thousand training data points (300 training points × 5 classes × 4 periods) for each province were collected for classification and change detection. Land cover changes at decadal and over the entire study period for five land cover classes were quantified at the country, provincial, and county level. High classification accuracy indicates localized land cover classification have an edge over the latest global land cover product and reveal fine differences in landscape composition. The vast steppe landscapes in these two countries are dominated by grasslands of 91.5% for Dornod in Mongolia and 74.7% for Aktobe in Kazakhstan during the 30-year study period. The most common land cover conversion was grassland to cropland. The cyclic land cover conversions between grassland and cropland reflect the impacts of the Soviet Union's largest reclamation campaign of the 20th century in Kazakhstan and the Atar-3 agriculture re-development in Mongolia. Kazakhstan experienced a higher rate of land cover change over a larger extent of land area than Mongolia. The spatial distribution of land use intensity indicates that land use hotspots are largely influenced by policy and its shifts. Future research based on these large-scale land use and land cover changes should be focused the corresponding ecosystem and society functions.

**Keywords:** LULCC; land cover classification; land use hotspots; landlocked country; Mongolia; Kazakhstan; Asia dryland; Google Earth Engine

## 1. Introduction

Kazakhstan and Mongolia, the two largest landlocked countries in the world, encompass a landmass of 4.3 million km$^2$ and host a combined population of 19.2 million people, amounting to some of the lowest population density in the world. Situated in the northwest of dual belts (i.e., the Asian dryland belt and the Eurasian steppe belt), this vast nomadic pastoral system faces unparalleled sustainability challenges, as these two countries have experienced relatively extreme climatic change, frequent shifts in institutions (e.g., changes

in land tenure policies), and hard economic transitions from state-controlled economies to free-market systems [1–6]. Kazakhstan was a member of the Union of Soviet Socialist Republics (USSR), whereas Mongolia was not an official member of USSR but has been strongly influenced by the USSR for its independency, administrative structure, defense, etc. since the 1920s. Both experienced catastrophic geopolitical disruptions in the 20th century with the dissolution of the USSR in December 1991. Since they gained independence, they have taken divergent routes of political reform and economic recovery [3]. The region's climate is predicted to have a warming trend and increasingly frequent extreme climate events [7,8]. Stressed by political shifts, intense climate change, and unstable economies, their social-environmental systems are under enormous pressure within the rigid water-limited environment. As a medium to the interdependent social, economic, and ecological systems, transformations in land cover reflect the interactions among these key elements of the environment and society [2,6,9].

Intensified land use has been identified as a major driver to changes in social–environmental systems (SES). The spatial and temporal changes of land use activities are very different in the two countries, due to their unique economic development stage and trajectory [2,3,10,11]. The most dramatic land use practice shifts happened during the second half of the 20th century, when their natural grassland ecosystems were converted to intensive agriculture [3,12–14]. Kazakhstan and Mongolia both have dedicated a large proportion of their land surface to food production at various times. During 1954–1956, the Virgin Lands Campaign transformed 13 million hectares of native steppe into wheat fields in northern Kazakhstan, turning it into the third-largest grain producer among the Soviet republics [15–17]. This campaign was also implemented in Mongolia but to a lesser extent. The largest agricultural development movement in Mongolia was called Atar ezemshik (Atar, for short, which means 'uncultivated land' in Mongolian), and its most influential phase of agriculture development, Atar-3, ran from 2005 until 2009, when the government financially supported reclaiming the abandoned agriculture lands [18]. Mining is another sector that posts huge potentials for major changes in land use. Both countries have vast untapped natural resources that could be extracted and will likely become mining hotspots in the region. In Mongolia, mining industry expansion has been coupled with a flux of migration from rural to urban centers [19], which converts natural vegetation to man-made impervious surfaces. According to Chen et al. (2022), using the Moderate Resolution Imaging Spectroradiometer (MODIS), the estimated land use land cover change (LULCC) during 2000–2020 was 4.7% and 5.3% for Kazakhstan and Mongolia, respectively, with the urban land cover increasing by 4.7% and 0.4%, respectively [3].

To better understand the pattern and impact of land use practice in Kazakhstan and Mongolia, we take a comparative approach to examine policy-driven land use changes by classifying land cover at fine spatial resolution and over a long temporal scale. Our goal is to investigate the extent and trajectory and the spatial variation of land cover change at three administrative levels (i.e., country, province, county) in the two countries and, ultimately, identify the land use hotspots. This investigation spans three decades, from 1990 to 2020, and covers some important geopolitical events and socio-economic changes. Our working hypothesis is that the direction and intensity of land cover change varies between the two countries due to differences in land use policy and policy shifts over time. We selected three provinces from each country that reflect a variety of the country's ecological and socio-economic conditions. We then compiled Landsat images of the six provinces into four decadal composites (1990, 2000, 2010 and 2020), classifying the land cover types into five classes (i.e., forest, grassland, cropland, barren and water). Decadal land cover change and overall land cover change were reported at three administrative levels. Land cover transitions and the intensity of land cover change were compared between the two countries and within their administrative levels. The spatial clusters of high land change intensity pixels were identified as the country's land use hotspots.

## 2. Materials and Methods

### 2.1. Study Area

Kazakhstan and Mongolia have similar biogeophysical conditions (e.g., dominancy of semi-arid/arid landscape and continental climate) and socioeconomic roots in the former Soviet Union. Yet, they took divergent routes of political reform and experienced different paces of recovery political shifts after the World War II, including the Virgin Lands Campaign (1954–1963) in Kazakhstan, Atar ezemshik (phase I starting in 1940, phase II in 1976, and phase III in 2008) in Mongolia, independence from Soviet Union in 1991, and joining the World Trade Organization (Mongolia in 1997 and Kazakhstan in 2015).

The names of the administrative divisions are different in the two countries. Kazakhstan is divided into regions (oblasts), which are further divided into districts (rayonys), whereas Mongolia is divided into provinces (aimags) and districts (soums). From this point forward, we will use the terms *provinces* and *counties* in reference to both countries.

Kazakhstan is the world's largest landlocked country, with a total area of 2.73 million km$^2$ and a population of 18.8 million. The population density is <7 people/km$^2$, making the country one of the lowest population densities in the world. The country encompasses an area as wide as 2930 km from the west to the east and 1545 km from the south to the north. With such a large span, latitudinal temperature and precipitation gradients foster a series of vegetation including forest, steppe, and desert. The three selected provinces (Aktobe, Akmola, and Almaty) are representative of these diverse landscapes (Figure 1). In particular, Aktobe is the second-largest region in Kazakhstan (after Karaganda to its south). It is rich in natural resources and is a large industrial region of the country. Mining and chemical industries drive its economic development. Akmola is located in central Kazakhstan and is one of the largest agricultural and livestock regions in the country. Almaty sits at the southeastern corner of the country and has been the capital city for many years. This province borders Kyrgyzstan in the south by Trans-Ili Alatau, a branch of the Tianshan mountains, and Xinjiang, China, in the northeast by Dzungarian Alatau. Its northwestern border runs along Lake Balkhash, whose basin drains primarily from Ili River—a significant transboundary waterway in the region.

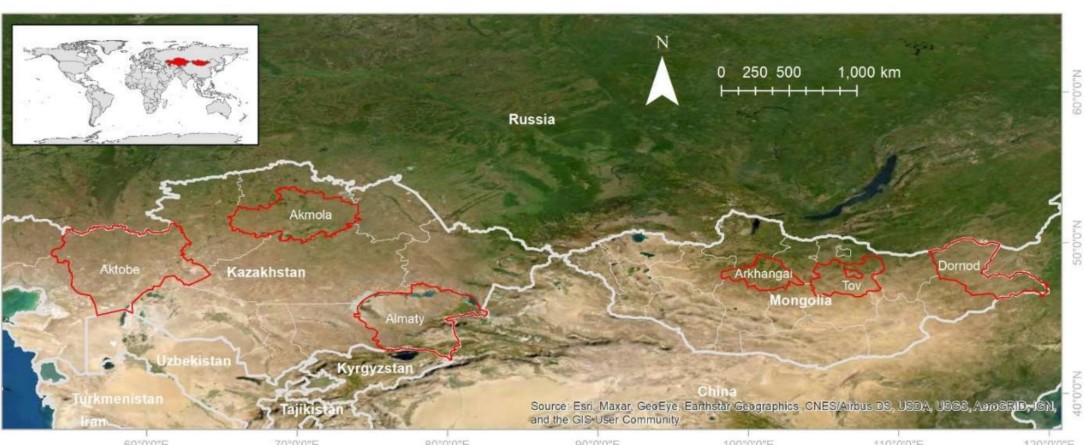

**Figure 1.** Positions of six provinces in two largest landlocked countries, including Aktobe, Akmola, and Almaty province in Kazakhstan, and Arkhangai, Tov, and Dornod in Mongolia which are highlighted by red line. The six provinces are selected based on their position along the climate, landscape composition, and socioeconomic gradients.

Mongolia, the second-largest landlocked country, covers 1.57 million km$^2$ (stretching 2392 km from east to west and 1259 km from north to south) and has a population of 3.3 million. A population density of ~2 people/km$^2$ makes it the least densely populated sovereign nation in the world. Its landscape is characterized by mountains mixed with steppes and vast plains and changes distinctly with latitude and longitude. From Mongol

Altai mountain (2500 m asl)—the highest mountain in Central Asia—the elevation drops eastward to Dornod plain at <700 m asl. The vegetation, ranging from Siberian taiga in the north to Gobi Desert in the south, is shaped by the climate gradient from cold and humid to warm and dry. The same gradient exists in mean annual precipitation: from 350–500 mm to <50 mm. We selected three provinces as our research sites: Arkhangai, Tov, and Dornod (Figure 1). Arkhangai received its name from the Khangai mountains which run northwest to southeast through the province. Mountainous landscapes dominate this province, where there is little arable land. Tov means 'central' in Mongolian, and the capital city (Ulaanbaatar) is in the center of Tov aimag. This province has the largest portion of cropland, which is needed to support the capital city. The Khentii mountains sit at the northeastern corner of Tov. Dornod means 'the East' in Mongolian, and Dornod is the easternmost province in the country, bordering China. Plain steppe landscape (i.e., relative flat) is the primary feature of Dornod province.

*2.2. Image Classification Procedures*

We leveraged the computational capability of Google Earth Engine (GEE) [20] to accommodate the immense processing of Landsat images for the vast study area. We applied pixel-based and supervised classification method to classify Landsat 5 TM, Landsat 7 ETM, and Landsat 8 OLI/TIRS (Tier 1 Surface Reflectance image collections) for the periods around 1990, 2000, 2010 and 2020. High clouds and limited spatial coverage for the study sites make obtaining a high-quality image composites from a single year nearly impossible, so we extended the time ranges to more than one year when compositing images. In brief, though each time period was considered the decadal year, we included images during and 1–2 years before and after the target year, as well as images of ±4 years in extreme cases, to ensure image quality. For example, Almaty did not have satellite coverage in 1986, and thin cloud images were not found prior to 1990, so we increased our range by using images from 1990 to 1994 to produce a mosaic that covered the entire province. As a result, the composite for Almaty in 1990 represents images from 1990–1994. Landsat collections of GEE were atmospherically corrected and georeferenced. For Landsat Surface Reflectance, there is a quality band named "Bitmask for QA_PIXEL" to filter snow, cloud and cloud shadow. Altogether, 6964 Landsat images were used in this study after screening.

Our image selections for each decadal year were drawn from those taken during summer months (1 June–31 August, or Julian day 152–243). For each image composite, blue, green, red, near-infrared, and shortwave infrared 1 and 2 were selected and computed to the 75th percentile. Vegetation indices, including normalized difference vegetation index (NDVI), normalized difference water index (NDWI), and visible atmospheric resistance index (VARI), as well as topographic variables (i.e., elevation, aspect, and slope) were included to form a composite of 12 bands for classification. Topographic data was acquired from an AW3D DSM elevation dataset (~30 m horizontal resolution) that is available on GEE platform [21]. User memory limits set by GEE and the large extent of research sites (particularly for provinces in Kazakhstan) limited the number of bands used for classification. The initial design included three seasonal composites (late-spring, mid-summer and early autumn), but Google Earth Engine has limited capacity for to process images, so we included bands according to the variable importance reported from random forest classifier on GEE.

We used the scheme of the International Geosphere-Biosphere Programme (IGBP) in defining land cover type, adapting it to the arid/semi-arid regions. The land cover types for this study include forest, grassland, cropland, barren, and water. Cloud and snow pixels were masked (snow pixels were only present in Tianshan mountains in the Almaty province, Kazakhstan). Urban was not considered in this classification, because, apart from the provincial capitals, the man-made structures (e.g., houses, roads) and other typical urban characteristics account for only a small portion of the landscape surrounding most cities in the two countries, especially in Mongolia. Residential houses and other buildings

in population centers often are scattered in the landscape, rendering the spectral signatures of urban lands indistinguishable from the surrounding dominant barren and grassland cover types at 30 m resolution. Integrating a night-time image and hierarchical model could improve the delineation of built-up areas, but not for our study landscapes where electricity was often not available [22]. Consequently, we excluded built-up cover type in our classification theme. Where urban boundaries were apparent for large cities, we excluded them from the image analysis if they met the following criteria: (1) population is ≥10,000 based on 2010 census, or (2) the city was a provincial capital. To make our classification consistent across the six provinces, we thus excluded 39 cities (35 from Kazakhstan and 4 from Mongolia) by masking them out from the image composites and classifications.

For each province region, we collected ~300 training data for each land cover type by allocating sampling points evenly across the landscape. We maintained a core set of training data for each nominal year and revisited the feature for land cover change. By doing so, we kept a consistent pool to train the random forest model for potential land cover changes. Altogether, we had ~6000 training data points (300 training points × 5 classes × 4 periods) for the six provinces. Random forest (RF) algorithm was used to train the classifier. RF function on GEE requires user to provide input with six arguments for a customized classification model, such as the number of trees, the number of variables per split, minimal leaf population, etc. Among these inputs, the number of trees can increases significantly overall classification accuracy [23]. We set this parameter to 30 while the remaining inputs were set with defaults. By adding a column of deterministic pseudorandom numbers to a feature collection (i.e., training features), the training samples were split into two groups for classification (70%) and accuracy assessment (30%). We also consulted local experts for qualitative validations. The landscape composition is disproportionate among the five land cover classes, with grassland and cropland dominate. Because we maintained a relatively large and balanced training dataset for each class and for each period (ca. 300 training points), this simple random sampling warrants an equal representation of the groups/land cover classes. Producer accuracy, consumer accuracy and overall accuracy of classifications are reported.

### 2.3. Land Cover Composition and Changes

The area and the proportion of each land cover class was tallied for each province and each time period. The default geographic coordinate system on GEE is WGS84. We project all images to Asia North Albers Equal Area Conic projected coordinate system when calculating areas of land cover classes within ArcMap. To visualize the land transformations between land cover types, an interactive Sankey diagram was generated to illustrate the conversions at decadal scale via R package OpenLand [24]. Both gross and net changes of each land cover type from 1990 to 2020 are also presented for each province using a modified OpenLand function.

### 2.4. Spatial Variations of Land Cover Change Intensity (iLCC)

To quantify the magnitude and intensity of land cover change (*iLCC*), intensity analysis was performed at both provincial and county levels. Intensity analysis computes the number of times a pixel changes during two periods [25]. Here, a pixel could change up to three times during the study period (i.e., 1990–2000, 2000–2010 and 2010–2020). First, *iLCC* was mapped at pixel level for each province, to identify the land use hotspots over the course of three decades. Second, the deviation (*iLCC—accumulative iLCC*) was calculated. This summarized the area of change by county and computed the percentage of area of change relative to county size for each province. *Accumulative iLCC* is essentially a zonal summary of *iLCC* by county. We compute *iLCC* with the exact_extract function in R package exactextractr [26] to shorten the computing time.

This summarization was presented in a bar graph and a choropleth map for each province. To find uniform class intervals for the two countries, a Jenks-style class interval

was applied for the continuous variable–accumulative *iLCC* via R package classInt [27]. This unified plotting color scheme is helpful to compare the variations of spatial change intensity among provinces and counties.

## 3. Results

### 3.1. Land Cover Classification and Accuracy

With the random forest algorithms and a large collection of training features, we achieved very high accuracies of land cover classifications as indicated by the three performance metrics (Table 1), due mostly to the large number of sampling points for each land cover class, and partly to metrics saturation, with an overall accuracy 0.9963. The consumer accuracy of the dominant land cover classes (0.9953 for grassland, 0.9962 for cropland) is lower than that of the minor land cover classes (0.9995 for forest, 0.9986 for barren, and 1.0 for water). The same pattern holds in producer accuracy except for barren land cover. The producer accuracy of barren is 0.9971, which is slightly lower than that of grassland (0.9972). We did not perform cross-validations with other land cover products because of the incompatibility in classification scheme.

**Table 1.** Consumer, producer and overall accuracies of land cover classification using Landsat images within the Google Earth Engine. The accuracy was averaged from all provinces and years.

| LCC Type | Consumer Accuracy | Producer Accuracy | Overall Accuracy |
|---|---|---|---|
| Forest | 0.9995 | 0.9995 | |
| Grassland | 0.9953 | 0.9972 | |
| Cropland | 0.9962 | 0.9936 | 0.9963 |
| Barren | 0.9986 | 0.9971 | |
| Water | 1.0000 | 1.0000 | |

### 3.2. Land Cover Composition

Land cover composition of the three provinces in Kazakhstan differed substantially (Table 2). As the second-largest province in Kazakhstan, Aktobe had the highest grassland amount, with an average grassland cover of 225,469 km$^2$ (74.7%) during the 30-year study period. The remaining 25.2% of the landscape was comprised of cropland (15.3%), barren (7.8%), forest (1.5%), and water (0.6%). In Akmola, cropland (48.9%) and grassland (44.5%) co-dominated the landscape (93%). For Almaty, grassland encompassed two-thirds of the region (64.5%); the forests that accounted for 13.9% of the landscape are found in the mountain areas in the northeast (the Dzungarian Alatau range along the border with China) and the south (the Tianshan range that borders Kyrgyzstan and China). Almaty also had the largest proportion of water among three provinces (7.8%), due to the presence of Lake Balk—one of the largest lakes in Kazakhstan. In Akmola, grassland (44.3%) and cropland (49.1%) dominated the landscape. The rest of landscape was comprised of water (3.2%), forest (2.1%) and barren (1.4%), which were scattered across the landscape. In Mongolia, grassland also dominated the landscapes. Located in the easternmost Mongolia, Dornod had grassland comprising 91.5% of the landscape (30-year average). A small fraction of forest (4.2%) was clustered in the western and eastern parts of the province (Figures A1 and A2). Small cropland patches appeared in Khalkhgol county. In Tov, grassland made up 66.4% of the total, and the taiga forest filled a large portion (23.6%) in the northeast of the province in branches of the Khan Kentii mountains. Cropland was found on the plains between mountain ridges in the northwest. In Arkhangai, grassland accounted for 64.1% and forest covered 24.6% of the landscape. Forests were found in the Khangai mountains, and cropland occupied the east and central parts of the province (Figures A1 and A2).

**Table 2.** Area (percentage) of land cover changes for the six provinces of Kazakhstan and Mongolia during the 1990s-2020s. The unit for area is km$^2$; the average values are calculated from the four time periods.

| Country | Province | Year | Forest | Grassland | Cropland | Barren | Water |
|---|---|---|---|---|---|---|---|
| Kazakhstan | Aktobe | 1990 | 6198.3 (2.1) | 218,428.6 (72.4) | 50,955.1 (16.9) | 23,521.8 (7.8) | 2518.5 (0.8) |
| | | 2000 | 4617.3 (1.5) | 231,513.2 (76.8) | 40,193.4 (13.3) | 23,059.6 (7.6) | 2238.8 (0.7) |
| | | 2010 | 2945.5 (1.0) | 227,007 (75.3) | 43,587.4 (14.5) | 26,533.9 (8.8) | 1561 (0.5) |
| | | 2020 | 4111.1 (1.4) | 224,929.2 (74.6) | 49,379.2 (16.4) | 21,759.7 (7.2) | 1455.6 (0.5) |
| | | Average | 4468.05 (1.5) | 225,469.5 (74.8) | 46,028.8 (15.3) | 23,718.8 (7.9) | 1943.5 (0.6) |
| | Akmola | 1990 | 4445.4 (3) | 70,996.8 (48.4) | 65,779.8 (44.8) | 1414.1 (1) | 4073.1 (2.8) |
| | | 2000 | 2727.9 (1.9) | 84,357.4 (57.5) | 54,297 (37) | 2494.4 (1.7) | 2832.5 (1.9) |
| | | 2010 | 4918.9 (3.4) | 66,730.8 (45.5) | 69,212.6 (47.2) | 2366.8 (1.6) | 3479.2 (2.4) |
| | | 2020 | 3022.2 (2.1) | 64,927.5 (44.3) | 71,980.9 (49.1) | 2091.8 (1.4) | 4685.9 (3.2) |
| | | Average | 3778.6 (2.6) | 71,753.1 (48.9) | 65,317.6 (44.5) | 2091.8 (1.4) | 3767.7 (2.6) |
| | Almaty | 1990 | 28,533.1 (13) | 137,584.2 (62.7) | 39,128.8 (17.8) | 1066.1 (0.5) | 13,224.4 (6) |
| | | 2000 | 28,027.6 (12.8) | 144,742.6 (66) | 30,000.2 (13.7) | 2984.1 (1.4) | 13,405.7 (6.1) |
| | | 2010 | 36,055.5 (16.5) | 138,967.1 (63.4) | 23,182.6 (10.6) | 1405 (0.6) | 19,527.2 (8.9) |
| | | 2020 | 29,828.6 (13.6) | 145,006.4 (66.2) | 20,712 (9.5) | 907.5 (0.4) | 22,683 (10.4) |
| | | Average | 30,611.2 (14.0) | 141,575.1 (64.6) | 28,255.9 (12.9) | 1590.7 (0.7) | 17,210.1 (7.8) |
| Mongolia | | Year | Forest | Grassland | Cropland | Barren | Water |
| | Arkhangai | 1990 | 15,472.7 (28) | 35,336.9 (63.8) | 3338.5 (6) | 388.6 (0.7) | 810.7 (1.5) |
| | | 2000 | 12,863.2 (23.2) | 36,720.1 (66.3) | 4869.8 (8.8) | 248.5 (0.4) | 645.4 (1.2) |
| | | 2010 | 13,669.5 (24.7) | 34,455.2 (62.3) | 5940.6 (10.7) | 531 (1) | 751.3 (1.4) |
| | | 2020 | 12,583.1 (22.7) | 35,452.2 (64.1) | 6414.6 (11.6) | 307.9 (0.6) | 589.9 (1.1) |
| | | Average | 13,647.1 (24.7) | 35,491.1 (64.1) | 5140.9 (9.3) | 369 (0.7) | 699.3 (1.3) |
| | Tov | 1990 | 16,832.6 (22.7) | 48,158.5 (65) | 7117.1 (9.6) | 804.2 (1.1) | 1161.7 (1.6) |
| | | 2000 | 18,037 (24.4) | 45,353.7 (61.2) | 9332 (12.6) | 753.3 (1) | 597.4 (0.8) |
| | | 2010 | 17,847.2 (24.1) | 50,882.5 (68.7) | 4642.8 (6.3) | 578.3 (0.8) | 121.1 (0.2) |
| | | 2020 | 17,168.9 (23.2) | 52,542.4 (70.9) | 2544.4 (3.4) | 764.5 (1) | 1051.6 (1.4) |
| | | Average | 13,647.1 (23.6) | 35,491.1 (66.5) | 5140.9 (7.9) | 369 (1.0) | 699.3 (1.0) |
| | Dornod | 1990 | 7225.9 (5.8) | 110,355.1 (89.3) | 1757.5 (1.4) | 818.7 (0.7) | 3449.7 (2.8) |
| | | 2000 | 5410.7 (4.4) | 113,494.6 (91.8) | 2237.8 (1.8) | 914.6 (0.7) | 1549.2 (1.3) |
| | | 2010 | 3913.4 (3.2) | 114,892.3 (93) | 1889.2 (1.5) | 1494.9 (1.2) | 1415.9 (1.1) |
| | | 2020 | 4386.4 (3.5) | 113,774.7 (92) | 2921.8 (2.4) | 973.7 (0.8) | 1549 (1.3) |
| | | Average | 5234.1 (4.2) | 113,129.2 (91.5) | 2201.6 (1.85) | 1050.5 (0.9) | 1990.9 (1.6) |

*3.3. Decadal Land Cover Changes*

There exist clear differences in LCC between and within the two countries, as well as by cover type and by study period. In Kazakhstan, a high rate of land cover changes appeared in the first two decades of the study period (1990–2000 and 2000–2010) (Figure 2). The largest amount of change happened in 2000–2010 in Akmola. Grassland decreased by 17,627 km$^2$ (−12%) during this decade, while cropland expanded 14,915 km$^2$ (10.2%). These classes also experienced enormous changes in the previous decade (1990–2000), but in opposite directions: grassland gained 13,360 km$^2$ (9.1%) while cropland cover shrank 11,483 km$^2$ (−7.8%). In Almaty, the largest amount of change happened during 1990–2000 for grassland and cropland. Grassland increased 7158 km$^2$ (3.4%) and cropland decreased 9129 km$^2$ (−4.1%). The second largest amount of change was during 2000–2010, when both grassland and cropland decreased by 5776 km$^2$ (−2.6%) and 6818 km$^2$ (−3.1%), respectively. Interestingly, forest expanded 8028 km$^2$ (3.7%) during this decade. For all three provinces, the most recent decade (2010–2020) experienced the least change, with changing rates (~2%) a magnitude lower than that in previous decades (10%). In Mongolia, similar variations in land transformations exist by province and study period (Figure 2). The largest amount of change happened during 2000–2010, with agriculture predominating in Tov. Here, grassland cover increased 5529 km$^2$ (7.5%) and cropland decreased 4689 km$^2$ (−6.3%). The next largest change was found during 1990–2000, when grassland lost 2805 km$^2$ (−3.8%), while cropland gained 2215 km$^2$ (3%). In Dornod land cover changes appeared minor in all three decades for all types, although the changes were smallest in the most recent decade. In Arkhangai, two large changes occurred in forest and grassland cover: forest cover lost 2609 km$^2$ (−4.7%) in 1990–2000, and area of grassland cover decreased 2265 km$^2$ (−4.1%) in 2000–2010. Area of cropland kept gaining, but in decreasing increments over the course of the three periods (+1531 km$^2$ in 1990–2000, +1071 km$^2$ in 2000–2010, +474 km$^2$ in 2010–2020).

For the two provinces experiencing greatest amount of change (Akmola and Tov), the Sankey diagram shows clear patterns in the transformations among the land cover types (Figure 3). The principal land cover conversion in these provinces was between grassland and cropland. In Tov, 5522 km$^2$ of grassland was converted to cropland from 1990 to 2000, 8549 km$^2$ of cropland was converted to grassland in the next decade (2000–2010), and 8304 km$^2$ of cropland was converted to grassland during 2010–2020. In Akmola, 12,957 km$^2$ of grassland was converted to cropland from 1990 to 2000, 15,060 km$^2$ of cropland was converted to grassland in the following decade (2000–2010), and 18,231 km$^2$ of grassland was converted to cropland in 2010–2020. Meanwhile, in Akmola, 24,396 km$^2$ of cropland was converted to grassland in the first decade, 29,728 km$^2$ of grassland converted to cropland in the second decade, and 15,355 km$^2$ of cropland was converted back to grassland during 2010–2020.

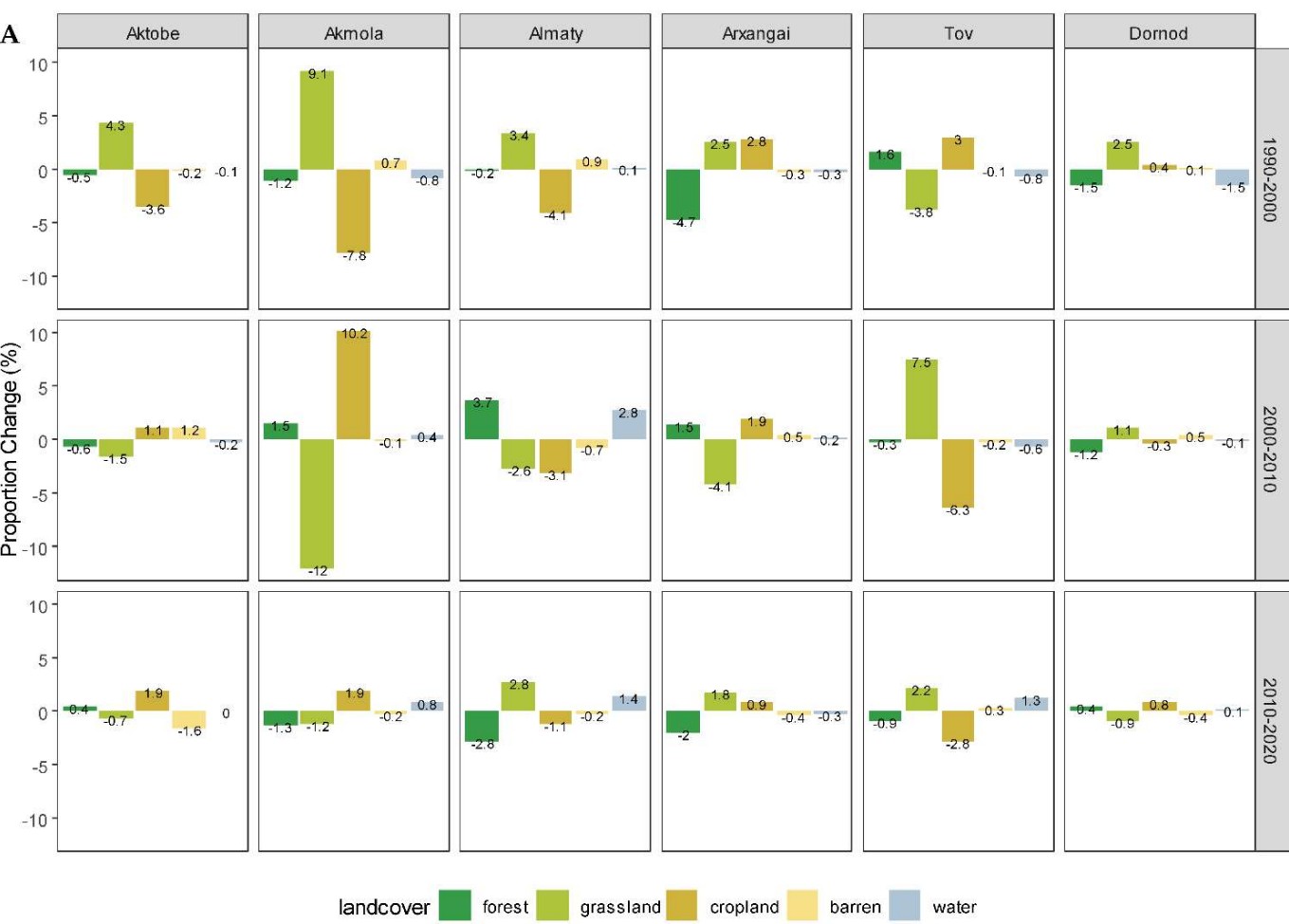

**Figure 2.** *Cont*.

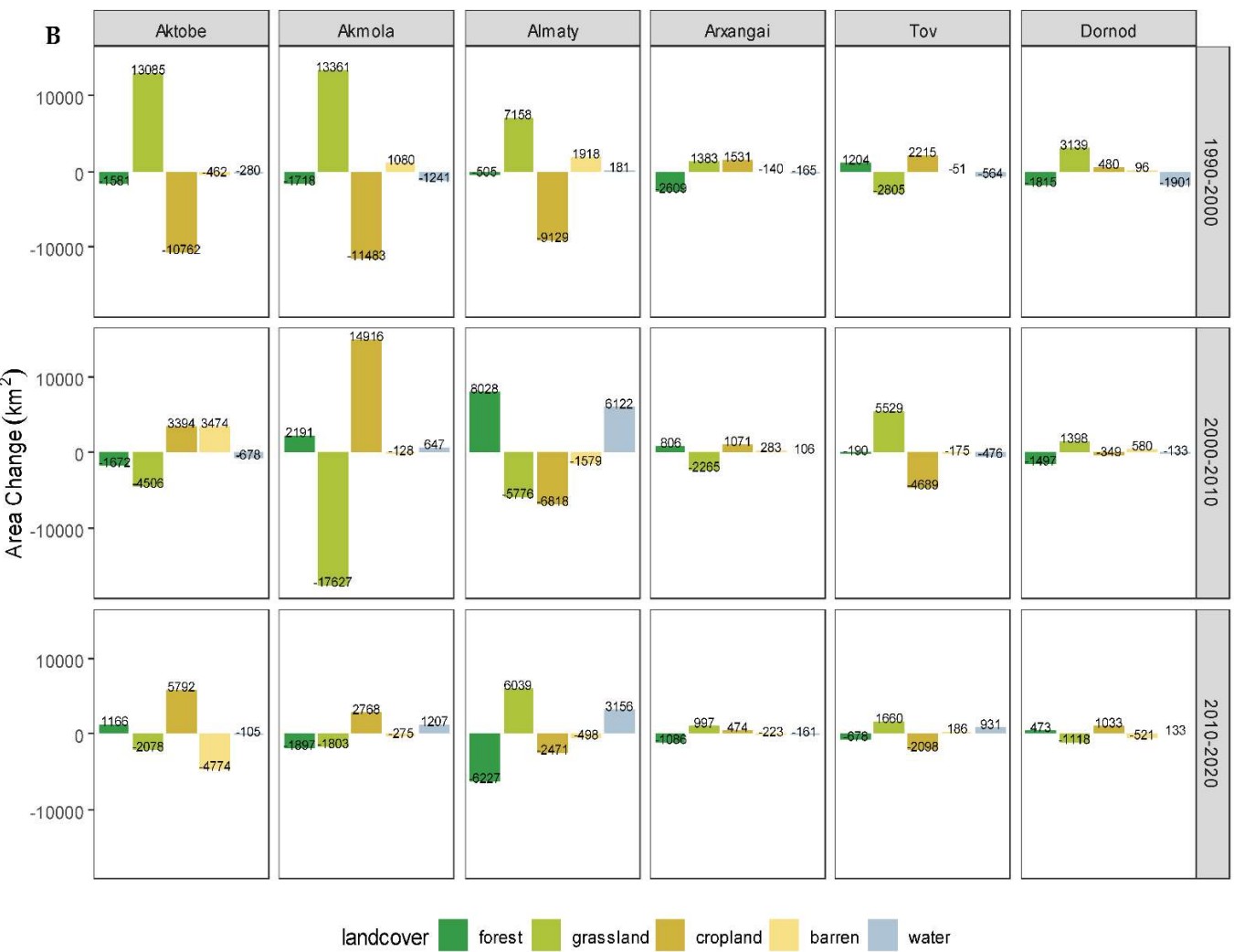

**Figure 2.** Decadal changes of land cover for the six provinces in Kazakhstan and Mongolia during 1990–2020. (**A**) Land cover change in proportion of the provincial total; (**B**) land cover change in area (km$^2$).

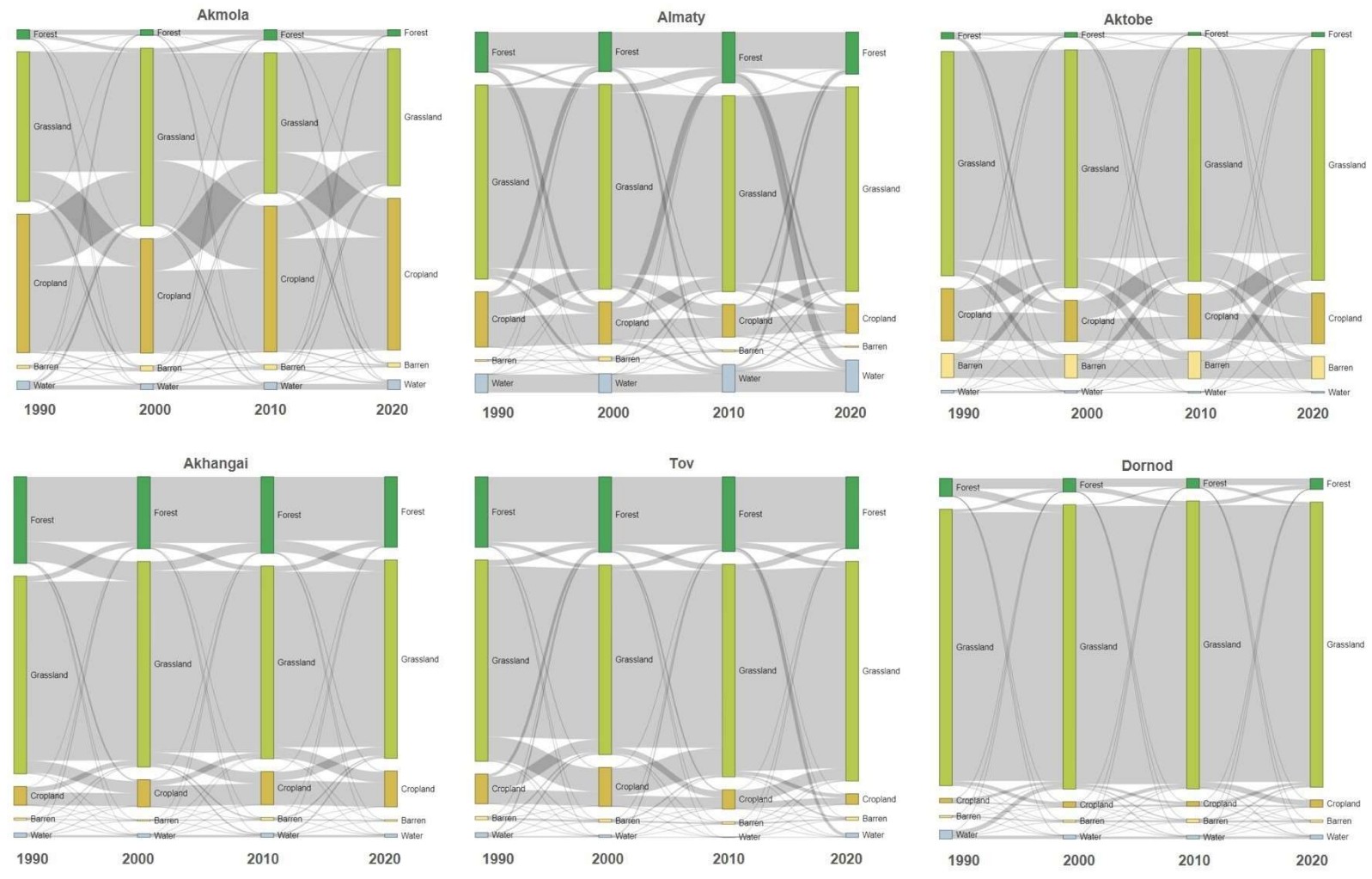

**Figure 3.** Sankey diagram of land cover conversions from 1990 to 2020 for six provinces. Land cover change in area (km$^2$). (Visit data section on project website for interactive figures http://lees.geo.msu.edu/research/FEW_MK.html accessed on 20 February 2022).

### 3.4. Gross and Net Change of Land Cover

Land cover conversions happened often and in large amounts during the three decades of the study period, with the net land cover changes providing another view. Despite the large amount of land cover change from decade to decade (Figure 2B), net gains and losses of land cover is small in three provinces of Kazakhstan (Figure 4). In Akmola grassland lost 6067 km$^2$ and cropland gained 6201 km$^2$. In Almaty, cropland experienced a large loss of 18,416 km$^2$, and grassland gained 7748 km$^2$. Forest contracted in Aktobe and Akmola in small amounts. Considering the small percentage of forest in the Aktobe and Akmola, this increase is trivial. In Mongolia, the net loss of cropland was 4572 km$^2$, and grassland gained 4384 km$^2$ in Tov. In Dornod, grassland and cropland gained but in different amounts, with a larger gain in grassland (3419 km$^2$) than cropland (1164 km$^2$). There was also a loss of forest cover (2839 km$^2$) over the three decades. In Arkhangai, cropland gained 3076 km$^2$ and forest loss 2890 km$^2$ over the course of thirty years. The large difference between gross and net change in land cover classes reflected the cyclical nature of certain types of change such as cropland abandonment and re-cultivation. Kazakhstan had a larger gross/net gain change than that of Mongolia, mostly attributed to Kazakhstan's larger size than Mongolia.

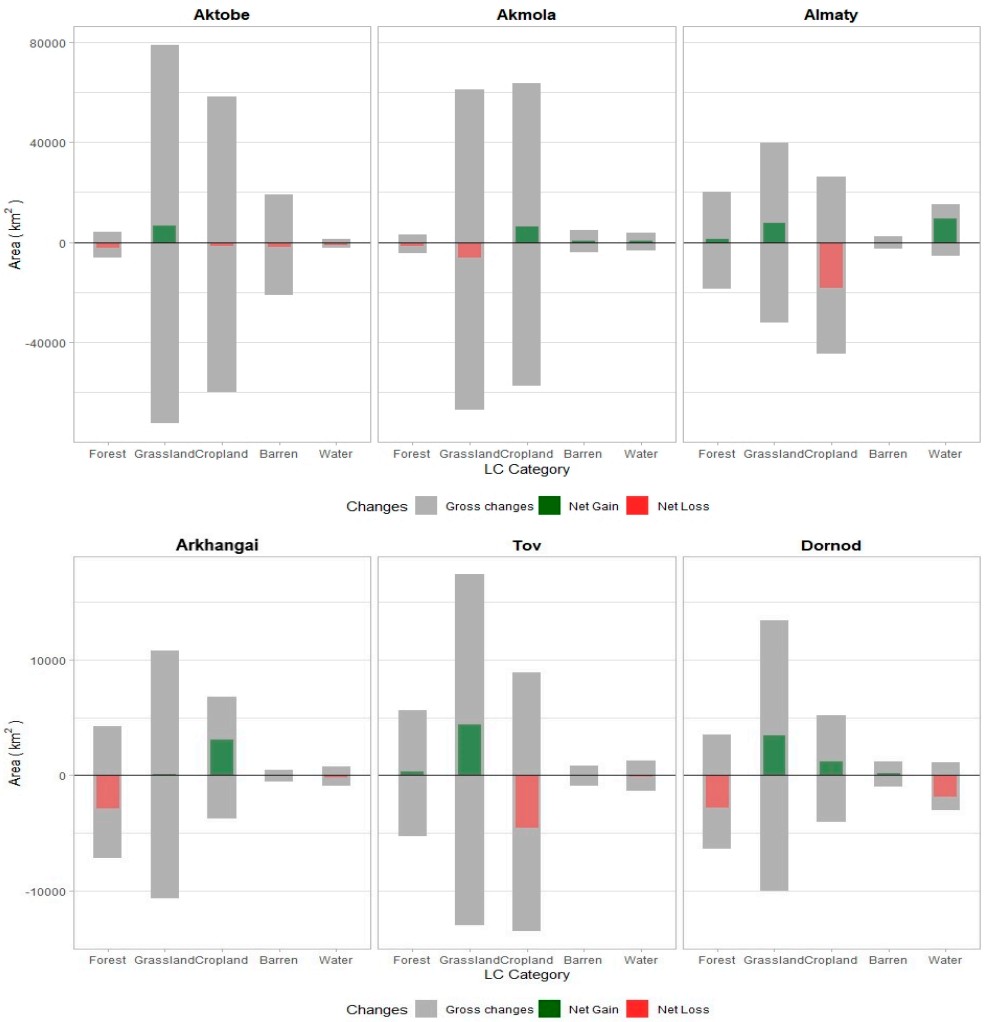

**Figure 4.** Gross and net changes of five land cover classes in six provinces in Kazakhstan and Mongolia from the 1990s to the 2020s.

### 3.5. Spatial Variations of iLCC

The intensity of land cover change (*iLCC*) in Kazakhstan was large (43.7% overall) but varied among provinces (Figure 4). The largest *iLCC* was found in Akmola (53.8% on

average), where 11 out of 17 counties experienced land cover changes above the provincial average (Figures 5 and 6). In Aktobe, land cover change was less intense, with an average of 48%, but maintained a relatively high percentage. All five counties with an overall change higher than the provincial average are clustered in the north, where cropland dominated the province. Although the overall spatial change for Almaty was 30.3% on average, a few counties experienced much larger changes. Counties in the south half of Almaty had a higher percent of change than the province average (30.3%). Overall spatial change was also evident in Mongolia but to a lesser degree (29.9% overall) compared to Kazakhstan. The largest changes occurred in Tov, with an average of 36.1%. Counties where *iLCC* was above the provincial average were found in the northwest mountain regions, where change was mostly related to agriculture expansion and abandonment. Counties in the south, where the terrain is flat and grassland was the dominant land cover, had relatively low change rates (<12.5%). Arkhangai experienced a large amount of change, with an average of 31.0%. There existed a distinct gradient from west to east (Figure 5). Spatial variations of land cover change intensity at county level for the six provinces in Kazakhstan and Mongolia. Jenks-style class intervals were used for the continuous variable (accumulative land cover change intensity, accumulative *iLCC*) toward a uniform color theme and to maximize the differences between classes. *iLCC* aligns with the topographic gradient well. Interestingly, the change rate appeared doubled, tripled and quadrupled from the west onward. The counties that had change rates greater than the provincial average sit at the eastern border. The province that experienced the lowest amount of change was Dornod (14.9% on average). Larger amounts of land cover change were found in the northwest mountain areas, with small amounts of change in the southeast flat terrain where grassland dominated the landscape.

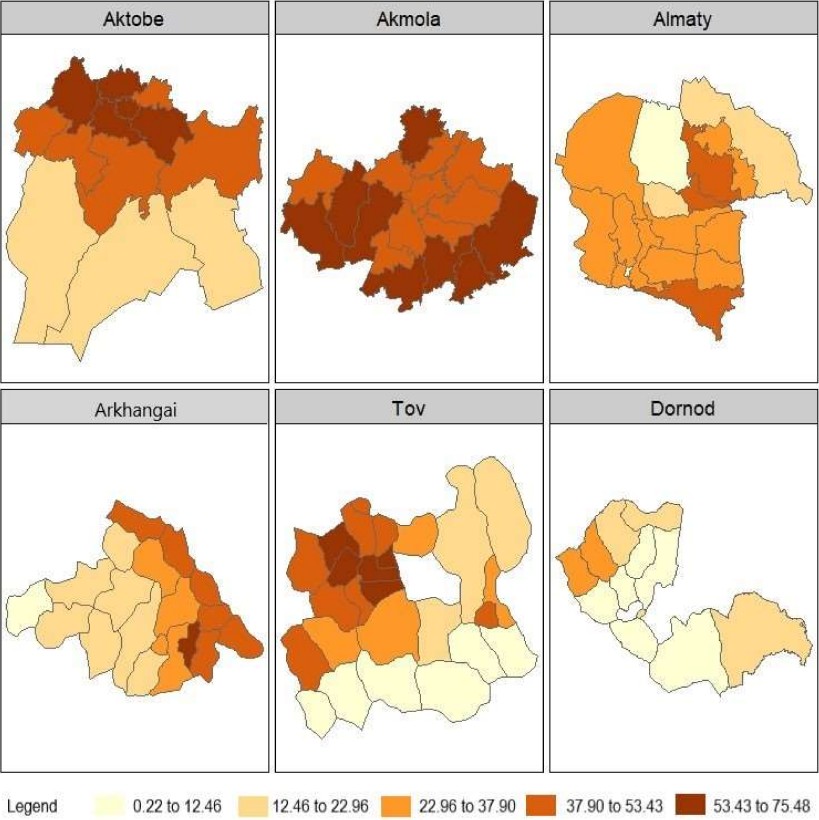

**Figure 5.** Spatial variations of land cover change intensity at county level for the six provinces in Kazakhstan and Mongolia. Jenks style class intervals were used for the continuous variable (accumulative land cover change intensity, accumulative *iLCC*) toward a uniform color theme and to maximize the differences between classes.

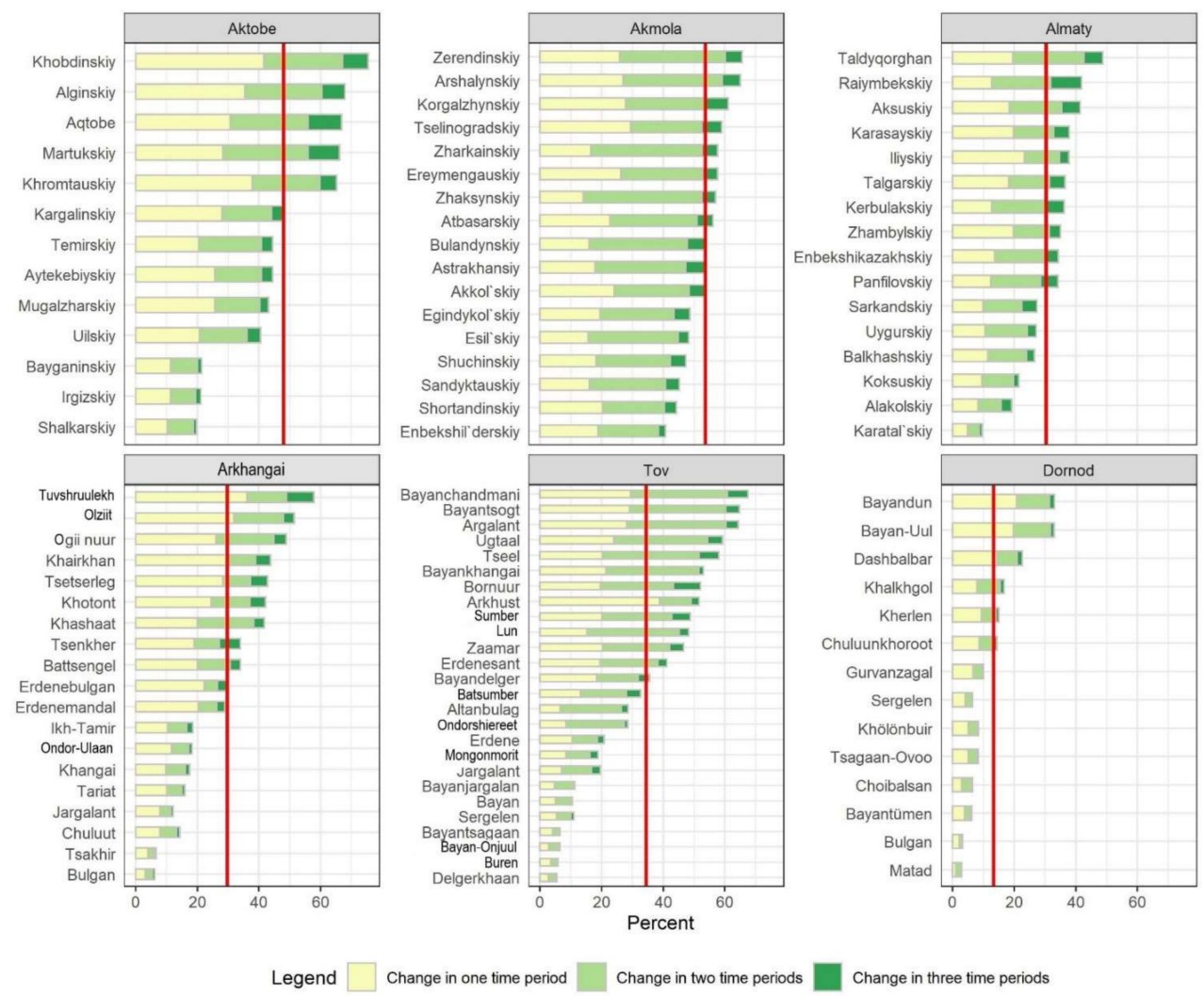

**Figure 6.** Accumulative land cover change intensity (accumulative *iLCC*) at county level by province over three decades (1990–2020). Vertical red lines denote the average of accumulative land cover change intensity of the province.

## 4. Discussion

### 4.1. Uncertainty in Land Cover Classification

Development of accurate land cover maps at finer resolution and over longer time spans is needed for more accurate understanding of local landscapes, as well as for improving global land cover products. The superiority of localized classification compared to global land cover products has been widely reported. This is especially true in Kazakhstan and Mongolia, where distinguishing cropland and grassland remains very challenging. For example, we found land cover and distributions remarkably improved at the same spatial resolution (30 m) in our classification product, compared with the latest global land cover product [28]. Gong et al. (2019) [28] found Tov province to be covered by grassland with a negligible amount of cropland, we found cropland to make up 7.9% of the landscape, a finding that is supported by national annual statistics (Figure 7A).

We nonetheless are aware of the limitations and uncertainty in our classification. First, excluding urban from our classification scheme hinders our understanding of the role that urban expansion played in provinces where it was the fastest-growing sector of the economy. This exclusion was necessary to avoid overestimating other land cover types In Kazakhstan and Mongolia. As described in the methods section, the characteristics of urban land cover, such as impervious surfaces, are not distinct in small and medium

towns, suggesting those lands would likely be classified as non-urban types (i.e., barren or grassland) [22,29]. An effort to identify these unique areas as urban would require different classification algorithms and remote sensing imageries (e.g., high resolution images from VENμS). In this study using Landsat imagery, leaving the urban class out appeared to be the best option. Secondly, distinguishing cropland from grassland has been challenging, resulting in high uncertainty regarding land cover conversion between the two. Giri et al. (2005) [30] reported that steppe vegetation ranges from forest to grassland to desert show gradual shifts (i.e., dynamic boundaries) between grassland and shrubland that can lead to misclassification and disagreement in land cover products. In this study, we observed some transitions between cropland and grassland by assessing the cover types around the edges of croplands (Figure 7), which may be attributable to our pixel-based image classification method. Across the study landscapes, we did not find similar 'all-or-none' pattern in cropland conversion to those of Sankey et al. (2018) who adopted an infusion of object-based and pixel-based image classification methods, as well as kernels to filter out misclassified land cover classes [31].

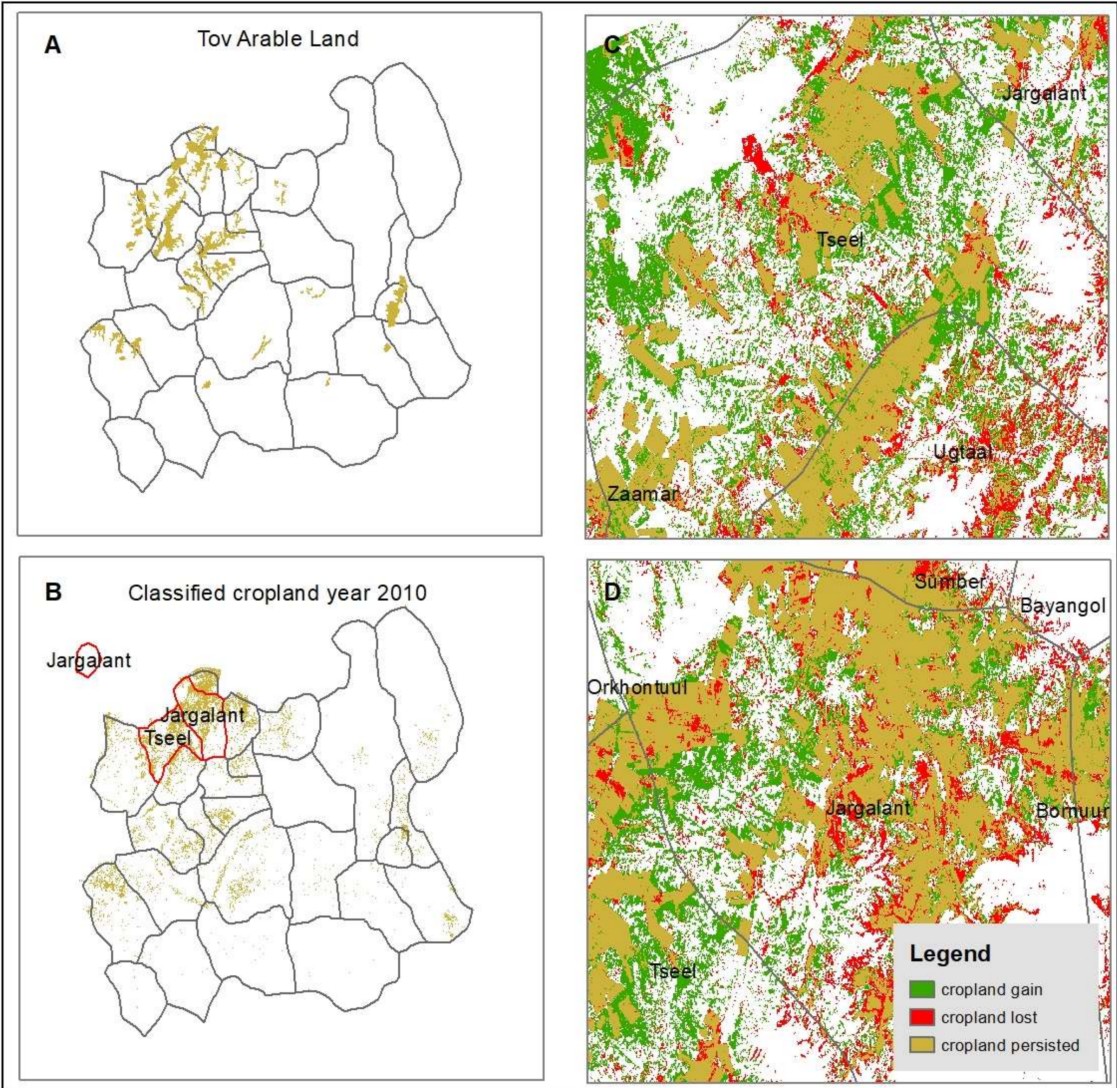

**Figure 7.** Spatial distribution of cropland. (**A**) Arable cropland distribution in Tov according to Mongolian government statistics; (**B**) classified cropland cover map of Tov in 2010 with highlighted counties having a large area of cropland; (**C**) cropland transition from 2000 to 2010 of county Tseel; and (**D**) cropland transition in Jargalant from 2000 to 2010.

Misclassification between grassland and cropland will likely under- or over-estimate the size of cropland in particular. For instance, a large amount of land cover conversion from cropland to grassland during 1990–2000 was found for three provinces in Kazakhstan (Figure 2) as a result of agriculture abandonment after the collapse of the USSR in 1991. The trend and extent of this decadal land cover change agree well with previous research [14,32], though the magnitude of the land transformations differed. Neither previous results nor ours match well with the ground surveys. In Kazakhstan, for example, it has been estimated that the grain production area decreased by 2 million hectares per year during the period 1993–1999 (i.e., 200,000 km$^2$ per year nationwide; USDA 2010 [33]). In this study, the loss rate of cropland in Akmola over a similar period (1990–2000) was 11,483 km$^2$ (i.e., 1148.3 km$^2$/year), which is very different from the estimates of de Beurs and Henebry (2004). Clearly, cropland remains significantly underestimated in this study. Regardless of high accuracy for all cover types, overestimates of some types or some places may exist as well. In Mongolia, the magnitude, direction, and scale of land cover changes do not align well with the governmental statistics. The National Statistics Office of Mongolia reported total arable land of 572 km$^2$ in Arkhangai, 2991 km$^2$ in Tov and 1278 km$^2$ in Dornod for 2020. Our estimates are an order of magnitude larger than the governmental statistics for Arkhangai (6414.6 km$^2$) and two times higher for Dornod (2921.8 km$^2$), albeit our estimate for Tov (2544.4 km$^2$) matched well. Worse yet, more misalignment was found for the direction of land cover change in Mongolia. The Atar-3 program, started in 2008, was a watershed for crop production in Mongolia. Prior to this initiative, croplands were distributed on lower slopes when the country changed from a planned economy to a market economy in the early 1990s. This was extended uphill after government-led, nationwide land Atar-3 cultivation campaign. However, these changes were not reflected in classified land cover products that indicate a slight increase in cropland area in all three provinces in 1990–2000, but a large decrease in Tov during 2000–2010.

*4.2. Land Use Hotspots and Policy Influences*

A nation's governance and policies can produce short- and long-term consequences in land use and land cover changes [3,19,34]. The intensive land cover conversions that occurred at decadal scale between cropland and grassland are closely related to institutional shifts in the two nations. As described earlier, the Virgin Lands Campaign (1954–1963) in Kazakhstan drove an influx of immigration, mostly from the former Soviet Union, into northern Kazakhstan, where large amounts of steppe were converted to cropland. This conversion was quickly reversed when the Soviet Union broke up in 1991, with large areas of cropland abandoned as farmers lost massive government subsidies [16]. The decline in grain production was accelerated in the mid-1990s when the livestock inventory shrank, leading to a further decline in the demand for feed-grain [35]. The land cover change trend found in this study (Figures 2 and 3) and others [32,36] for Akmola confirmed these changes, demonstrating the direct impacts of institutional shifts. After 2000, both crop and livestock production started to recover in Kazakhstan, due to governmental support for agriculture development [37]. Croplands were re-cultivated, resulting in major increases in cropland area (Figure 2B). Ironically, the conversion from grassland to cropland in last decade was reversed, resulting in minimal net changes in major cover types over the 30-year study period. Clearly, governance and institutional changes need to be factored in explorations of land cover changes as are other ecosystem and society functions [3,6].

At the provincial level, the direction and rate of land cover class change witnessed large divergences (i.e., within-country variations). It appeared that three provinces in Kazakhstan underwent a similar changing trend of increased grassland and decreased cropland area as a result of abandoned cropland reverted to grassland after the Virgin Lands Campaign in 1960s, when Kazakhstan then was a member of the Soviet Union. Among the provinces, Akmola experienced the highest changes, with its cropland cover decreased by 7.8%. In Mongolia grassland area expanded in Arkhangai and Dornod but not in Tov. Meanwhile, cropland in Tov expanded greatly, unlike in the provinces in Kazakhstan. At

county level, the spatial variation of land cover changes seemed more topography-driven. The highest land cover change intensity was found for croplands on flat terrains (i.e., more suitable for cultivation).

In Mongolia the influences of economic policies on land use hotspots were found at provincial level. For example, land use hotspots are clustered in the six counties in the northwest of Tov, mostly in Jargaland, Tseel, Ugtaal, Bayantsogt, Argalant, Bayankangai (Figures 7 and 8). These agriculture lands serve as the 'breadbasket' for the capital Ulaanbaatar. Another land use hotspot is located in southeastern Tov, mostly in Bayan and Bayanjargalan counties, where mining industries have been promoted by national policies since 2000 [19].

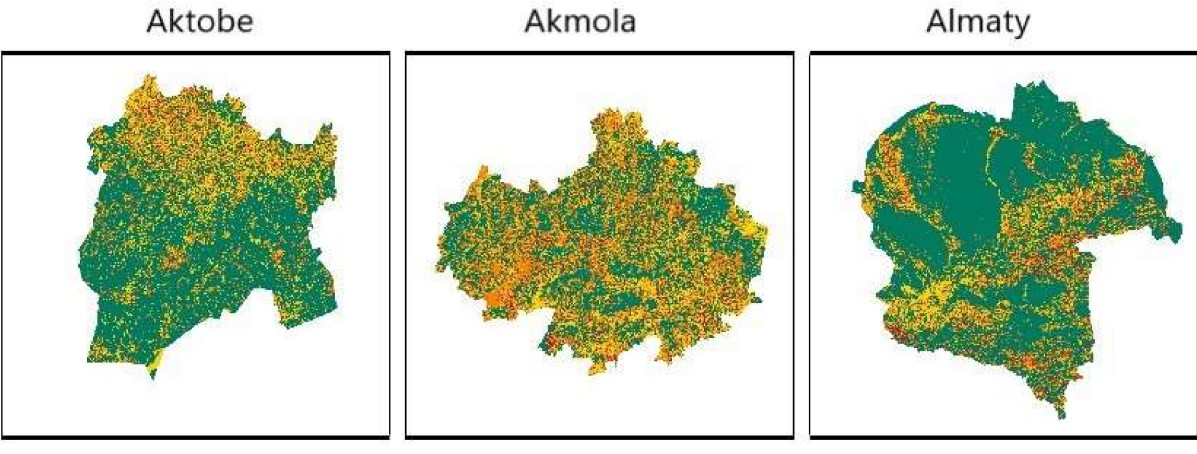

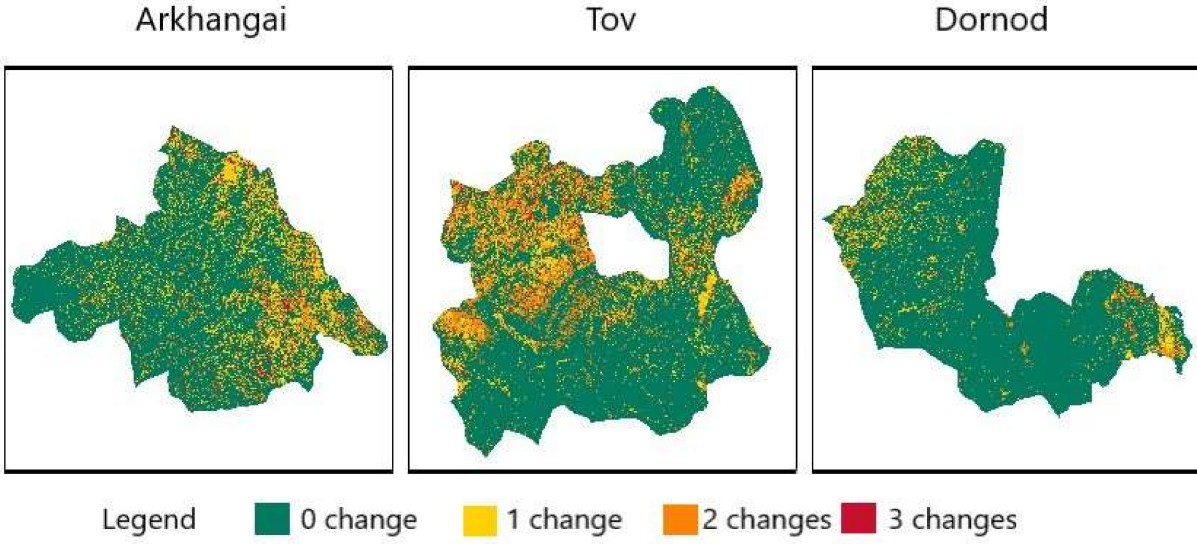

**Figure 8.** Hotspots of land cover change identified with *iLCC* which was computed the number of times a pixel that changed during the study period within a province. A pixel can change from one to three times during three time periods (i.e., 1990–2000, 2000–2010 and 2010–2020).

*4.3. Ecosystem and Climate Impacts*

With large scales and intensities of land surface alteration, one expects significant changes in ecosystems, societies, and the underlying regulations from climate and human forcings [38]. Many researchers have shown that changes in land use have remarkably influenced dryland ecosystem carbon cycling and sequestration via modifying the physical properties of the land surface (e.g., albedo, roughness, and evapotranspiration). In situ measurements of biomass in similar ecosystem in nearby Inner Mongolia, China, have shown that conversion of steppe into agricultural land leads to carbon loss, because grassland is higher in belowground net primary production than cropland, and most cropland biomass is returned to the atmosphere through harvest, and at the same time loses its carbon sequestration ability [4]. Landscape-scale investigation illustrates that vegetation degradation leads to albedo increasing by 5% and ET decreasing 0.8 mm/day [39]. Based on satellite image interpretation, Zhou et al. (2021) found a daytime cooling effect on crop dominated lands and a net daily cooling effect in arid zones based on MODIS land surface temperature [40]. If we multiply the per unit of carbon/albedo/ET change with the area of cropland being converted and reverted at the time scale, the numbers are overwhelmingly large and crucial on regional climate. As land use and land cover change is one of the most important disturbances that changes the terrestrial carbon pool and net flux at regional and global scales [41], we await answers to at least these pressing issues: What are the consequences of ecosystem carbon and water budget at landscape scale during land transformation? Will the directional land cover conversions result in similar changes in carbon, water and energy balances? What are the specific contributions of each change between cover type to the net change of a region or country?

We are aware of the age-old challenge in land change science: land cover is distinct from land use. Land cover addresses the layer of soil and biomass that cover the land surface, and is observable, while land use refers to land management practice and is not easily observable [42]. Just as Verburg et al. (2009, 2011) argued, an underlying distinction exists between land use and land cover, and more attention should be given to land use and land functions and linkages between these [42,43]. An example of the effect of this distinction is that changes in quantities of land area do not reflect differences in quality of land. The pervasive problem across Mongolia is overgrazing, which leads to land degradation and desertification [2]. It has been reported that 70% of the natural steppes in Mongolia were degraded under the pressure of climate change and overgrazing [44]. The land cover class 'grassland' does not show the degrading process and the discounted land function for provision of goods and services. The land use change as a result of grazing is not easily observable from satellite image interpretation. The variation in grassland management practice adds confusion to the classification model, which further leads to grassland misclassification. Another example of the effect of land use and land cover differences is the difficulty in documenting land abandonment [42]. In the cases of Mongolia and Kazakhstan, the problem is monitoring and detecting cropland abandonment. Because of mass land and sparse population, planting croplands in Mongolia is mostly opportunistic and based on funding availability of individual household. The Mongolian Statistical Office only records total harvest area annually [31]. This most likely leads to the discrepancy between the area of classified cropland and agricultural statistics that we discussed earlier. There have been research efforts to map cropland abandonment using satellite imagery in Mongolia [31], European Russia [45], and Kazakhstan [46], but the limitations in spatial and temporal coverage persist. The success of mapping cropland abandonment relies on developing a long time series of satellite images, which is computationally demanding and labor intensive due to the need of large training samples. Finally, land cover and land use shall be examined from functional perspectives, ecological and/or socially, so that meaningful lessons can be learned on their significances in shaping the nature and society.

## 5. Conclusions

The two largest landlocked countries, Kazakhstan and Mongolia, share similar biophysical features and both experienced the largest political shifts in the 20th century, but they have taken very different routes of political reform and economic recovery. The localized land cover classification is often superior to global land cover product and reveal fine differences in landscape composition. Large training features and cloud computing capacity that facilitate high overall classification accuracy across land cover classes and years enable us to detect land cover changes. Covering thirty years in temporal coverage change (1990–2020), our classification captures critical geopolitical events in modern history and enables the examination of the impact of those events on the landscape. Both countries experience higher rates of land cover changes in the first two decades of our study (1990–2000 and 2000–2010) than the latest decade (2010–2020), but with clear differences in LCC between and within the two countries, as well as by cover type and by study period. Two agriculture-dominated provinces that housed the national capital (Akmola and Tov) experienced the greatest amount of land cover change. The most common land cover conversion in the two countries was grassland to cropland. The cyclic land cover conversions between grassland and cropland reflect the impacts of the USSR's largest reclamation campaign in the 20th century in Kazakhstan and the Atar-3 agriculture re-development in Mongolia. Kazakhstan experienced a higher rate of land cover change over a larger extent of land area than Mongolia. The divergent natural resource management account for the spatial variation in land cover changes of three decades at three hierarchical administrative levels. The spatial distribution of land use intensity (Figures 5 and 6) indicates that land use hotspots (Figure 8) are largely influenced by policy and its shifts over time. Following up efforts are needed to examine the consequences of ecosystem and society functions from these large scale land use and land cover changes. To effectively translate our lessons for the society, governance and institutional changes need to be included in understanding the land cover changes.

**Author Contributions:** Conceptualization, J.Y. and J.C.; methodology, J.Y. and J.C.; software, J.Y. and P.S., V.K. and S.S.; validation, B.O.; formal analysis, J.Y.; writing—original draft preparation, J.Y.; writing—review and editing, J.Y., J.C., P.S., R.J., and B.O.; visualization, J.Y.; supervision, J.C.; funding acquisition, J.C. and R.J. All authors have read and agreed to the published version of the manuscript.

**Funding:** This research was funded by NASA LULCC Program, grant number 80NSSC20K0410.

**Data Availability Statement:** Decadal year satellite composites and classified land cover maps for the study sites can be downloaded from the project website hosted by Landscape Ecology and Ecosystem Science Lab (LEES) at Michigan State University. The link is: http://lees.geo.msu.edu/research/FEW_MK.html (accessed on 20 February 2022).

**Acknowledgments:** We appreciate the constructive suggestion from Likai Zhu on grassland land cover classification. Kristine Blakeslee of MSU Press provided careful editing of the manuscript. Three anonymous reviewers provided their valuable suggestions for improving the quality of this paper.

**Conflicts of Interest:** The authors declare no conflict of interest.

## Appendix A

**Table A1.** Counties with their urban areas removed from our classification.

| No. | Country | Province | City/County | Population |
|---|---|---|---|---|
| 1 | | | Shchuchinsk | 45,253 |
| 2 | | | Atbasar | 32,288 |
| 3 | | | Kokshetau | 146,104 |
| 4 | | Akmola | Makinsk | 18,540 |
| 5 | | | Ereymentau | 15,087 |
| 6 | | | Astana | 649,139 |
| 7 | | | Esil | 13,096 |
| 8 | | | Aktobe | 500,757 |
| 9 | | | Zhem/Embi | 12,345 |
| 10 | | Aktobe | Kndyagash | 25,553 |
| 11 | | | Alga | 15,372 |
| 12 | | | Shalkar | 26,329 |
| 13 | | | Khromatau | 24,089 |
| 14 | | | Almaty | 1,854,656 |
| 15 | | | Esik | 31,254 |
| 16 | | | Karabulak | 14,873 |
| 17 | | | Kaskelen | 37,221 |
| 18 | Kazakhstan | | Saryozek | 14,000 |
| 19 | | | Taldykorgan | 143,407 |
| 20 | | | Talgar | 43,353 |
| 21 | | | Tekeli | 31,958 |
| 22 | | | Usharal | 15,379 |
| 23 | | | Otegen Batyr | 17,301 |
| 24 | | | Sary-Ozek | 14,000 |
| 25 | | Almaty | Balpyk Bi | 12,145 |
| 26 | | | Uzynagash | 23,887 |
| 27 | | | Zharkent | 42,617 |
| 28 | | | Kargali | 20,114 |
| 29 | | | Koksu | 40,105 |
| 30 | | | Kapchagay | 33,428 |
| 31 | | | Zhansugirov | 8288 |
| 32 | | | Sarqan | 14,305 |
| 33 | | | Shelek | 26,688 |
| 34 | | | Saryozek | 14,000 |
| 35 | | | Ushtobe | 22,472 |
| 36 | | Arkhangai | Tsetserleg | 17,770 |
| 37 | | Dornod | Choibalsan city | 40,439 |
| 38 | Mongolia | | Ulaanbaatar | 1067,472 |
| 39 | | Tov | Zuunmod | 14,568 |

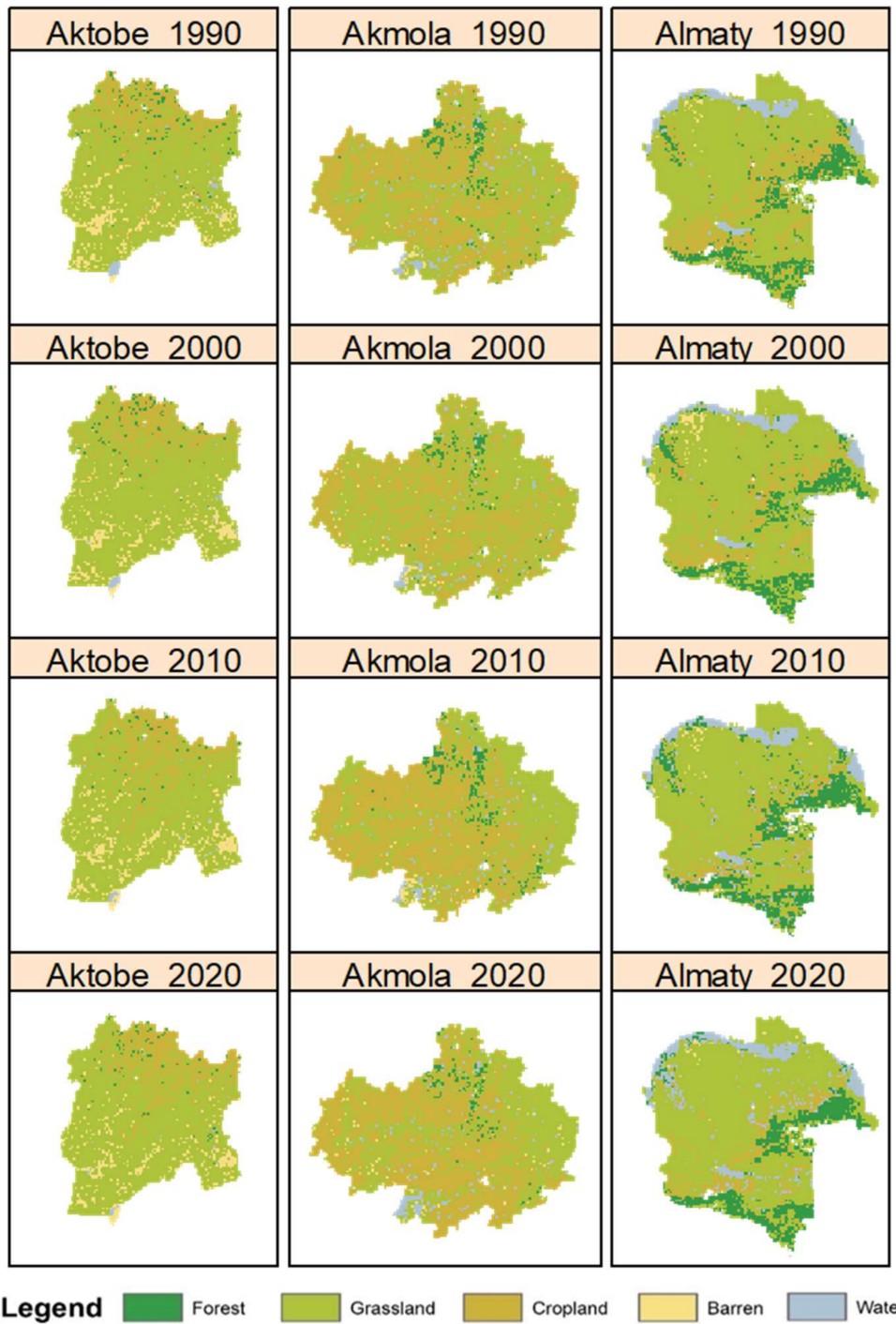

**Figure A1.** Classified land cover maps for the three provinces in Kazakhstan.

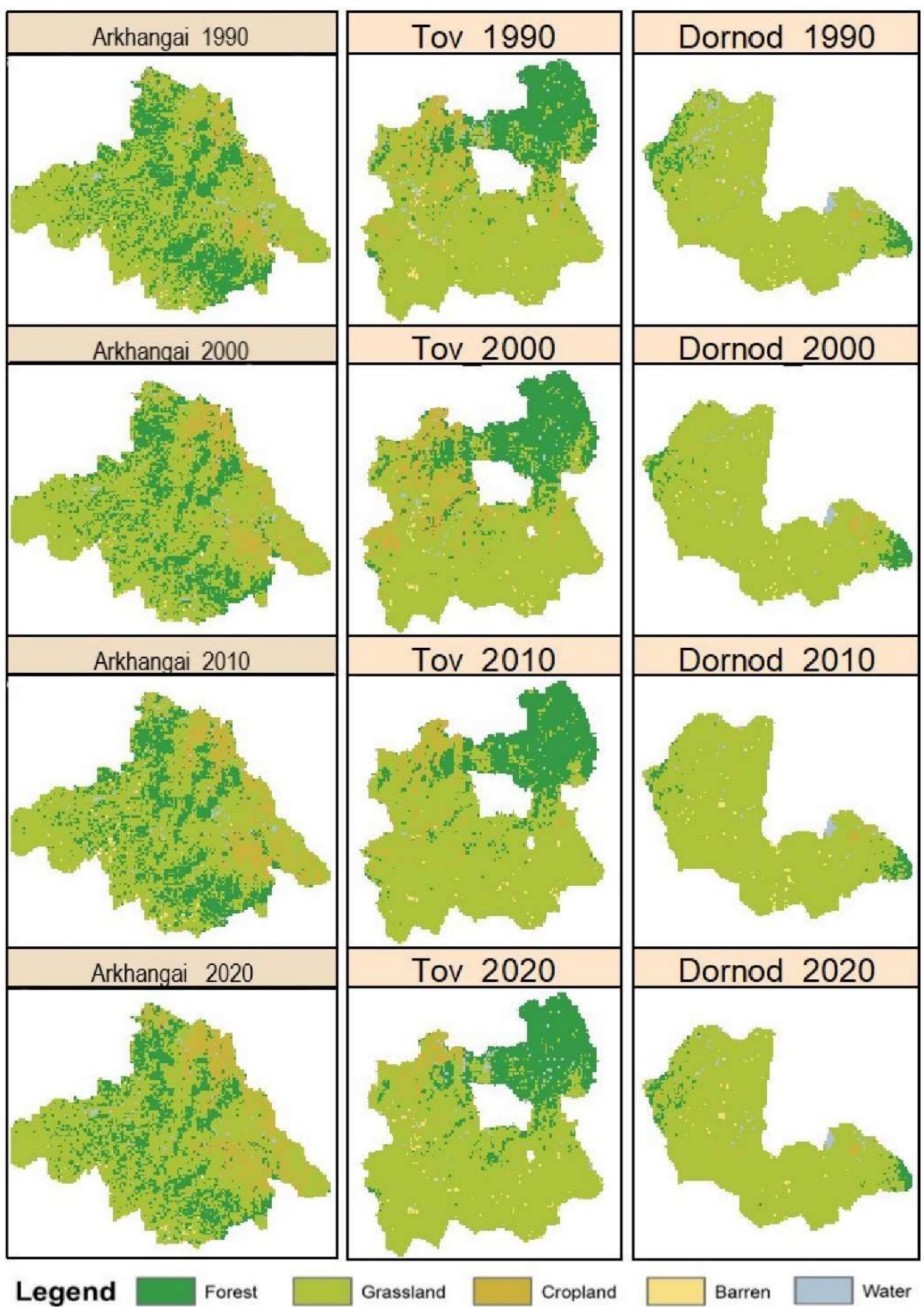

**Figure A2.** Classified land cover maps for the three provinces in Mongolia.

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
