# Peer review of "Land Use Hotspots of the Two Largest Landlocked Countries: Kazakhstan and Mongolia"

_remotesensing, doi:10.3390/rs14081805_

Round 1
Reviewer 1 Report
This paper presents an interesting study on land use hotspots of the two largest landlocked countries. The results look promising. The paper is well written. I recommend the paper to be published after addressing several minor issues as below.
The methodology is not very clear. Is it possible to add a block diagram?
I understand why the authors put all tables and figures at the end of this manuscript, however, that makes the paper reader unfriendly. Is it possible to add hyperlinks to Figures and Tables?
Page 4, Line 164: ‘a high-quality image composites’ may be ‘composite’
Page 15, Table 2: It would be better if the authors can adjust the font size so that it does not split into two lines.
Page 22, Figure 8: There are overlaps of the legend over the text. Please correct them.
Author Response
This paper presents an interesting study on land use hotspots of the two largest landlocked countries. The results look promising. The paper is well written. I recommend the paper to be published after addressing several minor issues as below.
The methodology is not very clear. Is it possible to add a block diagram?
We are not sure what “Block diagram” is. but surmise the reviewer for a flow chart on the classification method. If this is the case, we understand it is very common to include a workflow diagram on remote sensing manuscript. We nonetheless argue that since the method is well developed, and we perform the land cover classification by a standard procedure with atmospheric corrected and georeferenced Landsat images on GEE, it is not necessity to add additional figure to well-described texts. Further, we did add more texts to clarify the process. We included elaborations on image preprocessing, parameterization of random forest model, accuracy assessment, and geospatial analysis (mostly in methods section: Line 172-175,215-227, and 231-233). We hope these additions are at least equal to a flowchart.
I understand why the authors put all tables and figures at the end of this manuscript, however, that makes the paper reader unfriendly. Is it possible to add hyperlinks to Figures and Tables?
We apologize for the unsatisfactory reader experiences caused by not using the journal’s Word template. Remote Sensing journal accepts free format submission. In revised version, we reformated the manuscript based on the journal’s template, with all tables and figures embedded in the main text and near their first time uses.
Page 4, Line 164: ‘a high-quality image composites’ may be ‘composite’
We emphasize that it is a “high-quality” image composite because image with heavy clouds will be considered as “low quality”.
Page 15, Table 2: It would be better if the authors can adjust the font size so that it does not split into two lines.
Thanks for the suggestion! We tried in vain for a better presentation. We may leave this to the final proof when is accepted.
Page 22, Figure 8: There are overlaps of the legend over the text. Please correct them.
Correction was made. The overlapping in the text was due to the incompatibility between the Word template and journal template. We have reformatted the manuscript based on journal template and hope this will provide smoother reading experiences for reviewers.
Reviewer 2 Report
This manuscript is well written. People need this land use/land cover data for such landlocked countries.
Before acceptance, I have two concerns about the methods:
1) it seems that the training and validated data are not from ground-truthing, then how did the authors decide the random points to each land use type?
2) for these two countries, there are difficulties to classify the croplands from grasslands and barelands. The authors need to clarify this clearly, especially in how to separate them from each other.
For table 2, it is suggested to use 103km as the unit for the areas of each land-use type.
Author Response
This manuscript is well written. People need this land use/land cover data for such landlocked countries.
Before acceptance, I have two concerns about the methods:
1) it seems that the training and validated data are not from ground-truthing, then how did the authors decide the random points to each land use type?
The accuracy assessment for our product was performed within GEE. It adds a column of deterministic pseudorandom numbers to a collection (training features). It generated a random sampling on the training dataset by splitting them into two groups. Because we maintained a relatively large and balanced training dataset for each class and each period (ca. 300 training points), we are confident that each class has the similar probability as other classes to be selected. In addition, we consulted local experts from Mongolian Academy of Science and a co-author of this manuscript for qualitative validation. The stratified sample which the reviewer implies is helpful to generate an equal representation of the groups/land cover classes, but the fact this landscape composition are disproportionate for the five land cover classes, with grassland and cropland dominate, warrants the simple random sampling. We added clarification in Line 222-227.
2) for these two countries, there are difficulties to classify the croplands from grasslands and barelands. The authors need to clarify this clearly, especially in how to separate them from each other.
We understand these concerns. We feel we have provided ample discussions on the challenges in land cover classification in dryland environment. Additionally, we have included relevant references for readers for readers to explore further. Since our research focus is not on improving classification techniques but using land cover and land cover change as a medium to address the influence of political and socioeconomic shifts, we feel the volume on the background of remote sensing of dryland is adequate for this research.
For table 2, it is suggested to use 103km as the unit for the areas of each land-use type
Thanks for the suggestion! By convert the unit to 103 km2, it can certainly save lots of space and accommodate the large table 2 into one page, but we used km2 throughout the manuscript. In order to keep the units consistent, we believe it is better to have a consistent expression.
Reviewer 3 Report
The paper deals with the analysis of the land-use changes in Kazakhstan and Mongolia over a period of 30 years, which allowed highlighting the main land-use hotspots in the investigated countries by means of Google Earth Engine-aided processing of remotely sensed Landsat imagery. The results were mainly interpreted in the framework of political shifts and agricultural development movements. The topic is interesting, as land-use changes have, among others, deep impacts on land degradation and evolution of the ecosystems, and fits with the aims and scopes of Remote Sensing journal. The structure of the manuscript is good, and results are clearly separated from interpretations. The English is fluent. Figures are of good quality and help the reader to fully understand the results presented by Authors. I have only few concerns about the publication of the submitted manuscript.
- The manuscript, including the Reference list, must be formatted according to the “Instructions for Authors” of the Remote Sensing journal;
- Some more details should be provided about the methodology used by Authors. In particular: was it necessary the atmospheric correction of the used Landsat imagery? Which enhancement techniques were used? The Landsat imagery were already georeferenced? Which is the coordinate system? What about the extraction and the features of the land-use classes spectral signatures? Which algorithm was used for classification? About the validation statistics, I suggest reporting the confusion matrix and calculating sensitivity and specificity (Fielding & Bell, 1997; Environmental Conservation 24: 38–49; Beguerìa, 2006; International Journal of Remote Sensing 27: 4585–4598). Finally, how the spatial analysis was carried out? Did you use GIS software?
- Conclusions should not be a sort of second abstract. They should mainly report why the study was novel and which are the main open questions.
SPECIFIC COMMENTS
- Lines 12- 43. Abstract must be reduced to 200 words maximum, according to the journal guidelines.
- Line 32. Delete the parenthesis.
- Lines 57-58. To my knowledge, differently from Kazakhstan, Mongolia was never part of the Soviet Union, even if it was deeply influenced politically.
- Line 147: do you mean annual precipitation?
- Lines 266 and 271: Where is Figure S1?
- Lines 302, 305 and 307. I would avoid the verb "revert", as it is not sure that the areas converted to grasslands in the period 2000-2010 were the same that experienced the opposite land-use change in the previous decade.
Author Response
The paper deals with the analysis of the land-use changes in Kazakhstan and Mongolia over a period of 30 years, which allowed highlighting the main land-use hotspots in the investigated countries by means of Google Earth Engine-aided processing of remotely sensed Landsat imagery. The results were mainly interpreted in the framework of political shifts and agricultural development movements. The topic is interesting, as land-use changes have, among others, deep impacts on land degradation and evolution of the ecosystems, and fits with the aims and scopes of Remote Sensing journal. The structure of the manuscript is good, and results are clearly separated from interpretations. The English is fluent. Figures are of good quality and help the reader to fully understand the results presented by Authors. I have only few concerns about the publication of the submitted manuscript.
- The manuscript, including the Reference list, must be formatted according to the “Instructions for Authors” of the Remote Sensing journal;
We apologies for not using the journal’s template when we first submit. Remote Sensing states that it now accepts free format submission on its instruction for authors webpage. For revised revision, the manuscript was formatted using journal’s template. We hope this provide much easier reading experience for both reviewers and the readers.
- Some more details should be provided about the methodology used by Authors. In particular: was it necessary the atmospheric correction of the used Landsat imagery? Which enhancement techniques were used? The Landsat imagery were already georeferenced? Which is the coordinate system? What about the extraction and the features of the land-use classes spectral signatures? Which algorithm was used for classification? About the validation statistics, I suggest reporting the confusion matrix and calculating sensitivity and specificity (Fielding & Bell, 1997; Environmental Conservation 24: 38–49; Beguerìa, 2006; International Journal of Remote Sensing 27: 4585–4598). Finally, how the spatial analysis was carried out? Did you use GIS software?
To answer the reviewer’s questions:
- Landsat Images on GEE are atmospherically corrected and georeferenced. GEE provides description of each dataset. Here are the descriptions for Landsat 7 Level 2 Tier 1 which we used:
“This dataset contains atmospherically corrected surface reflectance and land surface temperature derived from the data produced by the Landsat 7 ETM+ sensor.”
From Specialized Algorithms session from GEE guides (https://developers.google.com/earth-engine/guides/landsat):
“Landsat surface reflectance (SR) data are available in Earth Engine as a copy of the USGS Collection 2, Level 2 archive. Note that Landsat 4, 5, and 7 SR data are generated using the LEDAPS algorithm, while Landsat 8 SR data are generated using the LaSRC algorithm.”
To better describe our image analysis procedure, we elaborated more on the technical details in lines 172-174: Landsat collections on GEE are atmospherically corrected and georeferenced. For Landsat SR, there is a quality band named “Bitmask for QA_PIXEL” which can be used to filter snow, cloud and cloud shadow. Also in Line 176-179 about clouds removal: For Landsat SR on GEE, there is a quality band “Bitmask for QA_PIXEL” which can be used to filter snow, cloud and cloud shadow. Considering GEE has endeavored to provide a ready-to-use collection.
- Coordinate: The default coordinate system of image is WGS84, a geographic coordinate system. We project all images to Asia North Albers Equal Area Conic projected coordinate system when calculating area of land cover classes in ArcMap. We add in line 237-240.
- Algorithm used for classification: It was mentioned in the manuscript, the classification used random forest model. We include more details on the parameterization on random forest model in line 220-221225: “RF function on GEE ask user to input six arguments for a customized classification model including the number of trees, the number of variables per split, minimal leaf population etc. Among them, increasing the number of trees significantly increases overall classification accuracy (Breisman, 2001). We set this variable to 30 and the rest of parameters are on default after rounds of experimenting”.
- Geospatial analysis: We include more details on our geospatial analysis in line 252-255. The calculation of iLCC is a zonal statistic using exact_extract from R package exactextractr instead of extract in R base package which slashes the computing time.
- Conclusions should not be a sort of second abstract. They should mainly report why the study was novel and which are the main open questions.
Thanks for the comments! We refined the conclusion to reflect the originality of our research.
SPECIFIC COMMENTS
- Lines 12- 43. Abstract must be reduced to 200 words maximum, according to the journal guidelines.
From the Instruction the journal provided on its website, it states “the abstract of about 300 words”. It is less than 300 words now.
- Line 32. Delete the parenthesis.
Deleted.
- Lines 57-58. To my knowledge, differently from Kazakhstan, Mongolia was never part of the Soviet Union, even if it was deeply influenced politically.
Yes, this is true. Mongolia was not an official member of USSR. But the country was mostly influenced by the USSR, including its administrative structure, defense, etc. Additional note is added in the manuscript in line 59-63.
- Line 147: do you mean annual precipitation?
Yes. Thanks for the suggestion. We made it clear in Line 145.
- Lines 266 and 271: Where is Figure S1?
It was at the end of the manuscript on page 24. Now it is in Appendix page.
- Lines 302, 305 and 307. I would avoid the verb "revert", as it is not sure that the areas converted to grasslands in the period 2000-2010 were the same that experienced the opposite land-use change in the previous decade.
Thanks for the suggestion! We replace “revert” to “convert” in line 331-338.
Round 2
Reviewer 3 Report
Dear Authors,
many thanks for your accurate and kind reply to my comments and suggestions.
In my opinion, the manuscript was significantly improved.
My recommendation to the Editor is accepting the paper in its present form, pending Editorial comments and final decision.
Kind regards.